# Contribution to a Circular Economy Model: From Lignocellulosic Wastes from the Extraction of Vegetable Oils to the Development of a New Composite

**DOI:** 10.3390/polym13142269

**Published:** 2021-07-10

**Authors:** Ivan Dominguez-Candela, Daniel Garcia-Garcia, Aina Perez-Nakai, Alejandro Lerma-Canto, Jaime Lora, Vicent Fombuena

**Affiliations:** 1Instituto de Seguridad Industrial, Radiofísica y Medioambiental (ISIRYM), Universitat Politècnica de València (UPV), Plaza Ferrándiz y Carbonell 1,03801 Alcoy, Spain; ivdocan@doctor.upv.es (I.D.-C.); jlora@iqn.upv.es (J.L.); 2Technological Institute of Materials (ITM), Universitat Politècnica de València (UPV), Plaza Ferrándiz y Carbonell 1, 03801 Alcoy, Spain; dagarga4@epsa.upv.es (D.G.-G.); aipena@epsa.upv.es (A.P.-N.); allercan@epsa.upv.es (A.L.-C.)

**Keywords:** chia seed flour, wood plastic composite, silane treatment, bio-polyethylene, circular economy

## Abstract

The present works focuses on the development of a novel fully bio-based composite using a bio-based high-density polyethylene (Bio-HDPE) obtained from sugar cane as matrix and a by-product of extraction of chia seed oil (CO) as filler, with the objective of achieving a circular economy model. The research aims to revalorize an ever-increasing waste stream produced by the growing interest in vegetable oils. From the technical point of view, the chia seed flour (CSF) was chemically modified using a silane treatment. This treatment provides a better interfacial adhesion as was evidenced by the mechanical and thermal properties as well as field emission scanning electron microscopy (FESEM). The effect of silane treatment on water uptake and disintegration rate was also studied. On the other hand, in a second stage, an optimization of the percentage of treated CSF used as filler was carried out by a complete series of mechanical, thermal, morphological, colour, water absorption and disintegration tests with the aim to evaluate the new composite developed using chia by-products. It is noteworthy as the disintegration rate increased with the addition of CSF filler, which leads to obtain a partially biodegradable wood plastic composite (WPC) and therefore, becoming more environmentally friendly.

## 1. Introduction

A global polymer production of 368 million tons was recorded in 2019. In Europe 58 million tons were produced and almost 25% of plastic post-consumer wastes is directly deposited in landfills [1]. The majority of conventional polymers are manufactured from fossil resources and are non-biodegradable. The most used polymers are polypropylene (PP), high- and low-density polyethylene (HDPE and LDPE) and polyvinylchloride (PVC), which represent some 60% of the plastics used [1,2]. Due to the mismanagement of these plastic products, most of which are single-use products, they can end up in landfills, oceans or other terrestrial ecosystems where they can affect wildlife and probably human health [3]. The use of biopolymers could be an excellent proposal for the plastics industry in order to overcome these drawbacks.

In recent years, biopolymers are gaining importance as a sustainable alternative to conventional polymers. They provide a 65% energy savings as well as between 30% and 80% less greenhouses gases emissions during their production compared to conventional polymers [4]. A biopolymer material is defined as a polymer that either is biodegradable, bio-based or has both properties [5]. This definition provides for three different types of biopolymers: (1) from renewable resources and biodegradable such as poly(lactic acid) (PLA) or polyhydroxyalkanoates (PHA); (2) from renewable resources and non-biodegradable, such as biopolyethylene (bio-PE) or biopolypropylene (bio-PP); (3) from fossil fuels and biodegradable such as polybutylene succinate (PBS) or poly(ε-caprolactone) (PCL) [6]. However, despite the great progress in biopolymers, these are still expensive, and, in some cases, present lower mechanical properties compared to conventional polymers. A growing alternative is the use of vegetable oils (VOs) that can be used as a raw material to obtain new products due to their ready availability and relative low cost. VOs present biodegradability and low toxicity, thus making them an attractive proposal for the plastic industry [7]. In addition, VOs and fatty acids from VOs, are considered one of the most important feedstocks to produce polymers materials and bio-based functional polymers [8,9]. The production of vegetable oils reached 210 million tons in 2020, which represents an increase of 25% in the last 8 years [10]. This production is mainly targeted to two sectors: human consumption and the chemical industry. Regarding human consumption, palm oil and soybean oil are the most consumed VOs, representing a 64% of worldwide production [10]. Moreover, VOs, which are mainly formed by triglycerides, can be chemically modified for the polymer industry due to the reactivity of the double bonds present in their structure. Commercially, there are non-modified VOs such as linseed oil or rapeseed oil used to protect the surface of woods [11], and modified VOs such as soybean and linseed oil used as a plasticizer in thermoplastics [7,12,13,14,15]. This increment of VO production is generating a large volume of residual cake as a by-product of oil extraction from whole seeds. Approximately more than 60% of entire seed is residual cake [16,17,18], a large volume of 524 million tons was produced in 2020. This residual cake, which is a lignocellulosic residue, is usually used as dietary supplement for livestock feed [19]. Another possible propose is reuse of the by-product as a lignocellulosic filler in the polymer industry contributing to the circular economy.

Wood plastic composites (WPCs), which consist of the addition of natural fillers in a polymeric matrix, are gaining more importance in order to reuse wastes from different industries. Some compounds from vegetable oil industry wastes such as linseed [20], palm [21], sunflower [22] or rapeseed [23] have been reported as lignocellulosic fillers in WPCs. This movement contributes to the circular economy, where residual cake rises in value as a filler to manufacture a sustainable composite. WPCs present attractive advantages such as low prices, excellent balance between mechanical properties, lightness, etc., [24]. Besides, they are eco-friendly, renewable, partially or completely biodegradable and they also lead to a decrease of the greenhouse effects [25]. The final properties of WPCs depend on the matrix, filler species and content, and the coupling agent used, among other aspects [25,26]. Another advantage of WPCs is that they can be applied as possible substitutes for natural wood due to their similar appearance. Several types can be used in furniture, fences, garden objects and so on [27]. However, although WPC have been widely studied, no evidence has been found previously or at least it is not available in the literature on the introduction of chia wastes from the vegetable oil industry as a lignocellulosic filler in polymer matrices.

Chia seed is commonly known for representing the highest food source of α-linolenic acid (omega-3) as well as a high nutritional content [28]. The current interest has led to a worldwide sales volume of some 90,000 metric tons for whole chia seed in 2017 [29] which represents the largest chia seed product market share at 45% [30]. Regarding chia seed oil (CO), it is becoming more widely used as a cooking oil, to manufacture health food supplements and in cosmetic products [31]. In addition, the high antioxidant capacity of CO has been employed to provide long shelf life in food products [32]. In 2018, CO accounted with market share of 20%, and is projected to present an annual growth rate of 23.4% from 2019 to 2025 [30]. Furthermore, modified CO could be suitable to use in the polymer industry due to its elevated unsaturated fatty acid levels, indicated by an iodine value of 190 [33]. In a previous work, epoxidized chia seed oil was obtained and applied successfully as a bio-plasticizer in a polymer matrix [34]. With the constant increment of CO production for the mentioned applications and taking into account that the CO extraction yield is around 25–30%, a percentage of residual cake around 70% of whole chia seed is produced after oil extraction. Therefore, the introduction of these lignocellulosic fillers in a polymer matrix to obtain a WPC could be an attractive proposal to reuse the by-products from CO extraction.

Traditionally, WPCs have been widely studied with conventional polymer matrices of petrochemical origin such as HDPE [24], LDPE [35], PP [22], PVC [36] and PS [37]. The growing tendency to manufacture green composites has increased the use of biopolymers to reduce greenhouse gas emissions. As an alternative, bio-based high-density polyethylene (Bio-HDPE) could be an interesting green substitute to the polymers of petrochemical origin. Bio-HDPE is obtained by conventional polymerization of ethylene obtained from catalytic dehydration of bioethanol which comes from natural resources [38]. One of the main advantages is that it presents the same physical properties, especially high ductility, and resistance, compared to polyethylene from petrochemical resources [39]. Besides, it presents easy processability to obtain injected pieces, which makes it suitable to industrial scale [40]. However, one of the main disadvantages is its non-biodegradability, although this can be partially resolved by adding lignocellulosic fillers which are biodegradable, thus manufacturing a WPC.

Given the current situation in which the use of vegetable oils in the polymer and food industry is attracting great interest, it is necessary to provide technical solutions to the possible waste generated in order to find a solution that is based on the circular economy. Therefore, the main objective of the present work was the reuse of the lignocellulosic waste obtained after the extraction of chia seed oil (*Salvia hispanica* L.) in a polymeric matrix based on Bio-HDPE. The study of compatibilization as well as the optimization of the lignocellulosic filler percentage is the polymeric matrix were carried out. A complete characterization of the composites has been carried out by means of mechanical (tensile, flexural, Charpy Impact Test or Shore D Hardness), thermal (DSC, TGA), thermo-mechanical (DTMA) and morphological techniques (Field Emission Scanning Electron Microscopy or FESEM) as well as water uptake, colourimetry and disintegration tests.

## 2. Materials and Methods

This paper has been divided in two parts. In the first part, the influence of a silane coupling agent has been determined, through the comparison of treated and untreated Bio-HDPE with 20 wt.% chia seed flour (CSF). In the second stage, an optimization of percentage of filler has been carried out in samples with 10, 20, 30 and 40 wt.% of treated CSF with silane coupling agent.

### 2.1. Materials

Commercial bio-based high-density polyethylene (Bio-HDPE) HA7260 grade has been employed as polymer matrix and was supplied by Braskem (Sao Paulo, Brazil). This polymer presents a minimum bio-based content of 94%, being obtained from bioethanol derived from sugarcane. Bio-HDPE present a density of 0.955 g·cm^−3^ and melt flow index 20 g·10 min^−1^ at 20 °C. As a lignocellulosic filler a residual cake obtained after cold pressed extraction of chia seed oil from entire chia seed (CS) supplied by Frutoseco (Bigastro, Alicante, Spain). was used. The residual cake was milled using an ultra-centrifugal mill from Retsch GmbH (Düsseldorf, Germany) at 8000 rpm and equipped with 0.25 mm sieve. After milling, CSF was obtained for use as a lignocellulosic filler. As silane coupling agent 3-(2-aminoethylamine) propyl]trimethoxysilane (APS) provided by Sigma Aldrich (Madrid, Spain) was employed.

### 2.2. Silane Treatment

Firstly, the selection of the coupling agent must be detailed. It is known that due to the highly nonpolar nature of the Bio-HDPE and the polar nature of the lignocellulosic filler, CSF, the use of compatiblizers is needed. Different options are reported in the previous literature: alkaline or esterification treatment of the lignocellulosic filler, use of titanate, zirconate or silane coupling agents or copolymerization with active molecules like maleic anhydride or acrylic acid, for example [41,42,43,44]. Silanization is one of the most employed methods to improve the adhesion/interaction between a polymer matrix and a lignocellulosic fiber. In previous works, APS has been compared with other silanes (glycidyl silane) and maleic anhydride, obtaining the best resistant mechanical and thermal properties [45]. In other studies, APS was compared with titanate and zirconate coupling agents in biocomposites developed with Bio-HDPE and eggshells. APS gave very similar results compared to titanate and better ones than zirconate treatment, despite the fact that the load introduced was calcium carbonate and not lignocellulosic. In addition, in previous studies, APS was used as a coupling agent in WPCs developed with Bio-HDPE, demonstrating an improvement of the interaction between fiber and polymeric matrix. Jordá-Vilaplana et al. used APS to improve the interaction between Bio-HDPE thermoplastic matrix and short fibers of *Cortaderia selloana* (Pampa grass) [46]. On the other hand, Carbonell et al. compared the use of APS and a polyethylene graft-maleic anydride copolymer in biocomposites developed with slate fiber and Bio-HDPE [47]. The results showed as APS leads to higher fiber-matrix interactions with positive effects on overall mechanical properties.

In the present study the silane treatment was carried out following the method reported by Fombuena et al. [48]. Briefly, 1 wt.% of APS with respect to filler weight was dissolved in water/acetone (50/50 *v*/*v*) to hydrolyze the silane to a silanol. The use of different solvents such as methanol or acetone has been broadly studied, but Pickering et al. reported that acetone improves the roughness of the lignocellullosic fillers, resulting in a large specific area [49]. Immediately, the pH of the solution was adjusted to 3.5 using acetic acid and the mixture was stirred for 15 min to ensure homogenization. The acid conditions improve the silane to silanol hydrolysis as well as slow down the self-condensation, which consists of the reaction between silanols forming polysiloxane structures [50]. The formation of polysiloxanes may hinder the diffusion of silanols into the lignocellulosic filler during the adsorption step [51]. In the following step, CSF were immersed in the previously obtained solution and mechanically stirred for 15 min. Finally, the CSF treated with amine silane was dried in an oven at 40 °C for 24 h to remove all residual solvents. In this step silanol molecules are adsorbed onto the lignocellulosic filler, acting as a chemical bridge in the interface after drying which allows to form chemical Si-O-C bonds between silanol groups and the hydroxyl groups presents in cellulose [52].

### 2.3. Sample Preparation

Prior to manufacture, untreated (UTCSF) and treated chia seed flour (TCSF) with APS were dried in an air oven at 50 °C during 24 h to remove the residual moisture. In the first stage, Bio-HDPE with 20 wt.% of UTCSF and TCSF were processed. In the second stage, five samples with compositions (by weight) of 0, 10, 20, 30 and 40% of TCSF were manufactured. All samples were weighed following the proportions of the Table 1. Compounds were hand mixed using a zip bag to homogenize them. The mixtures were introduced into a twin-screw extruder from DUPRA S.L. (Castalla, Alicante, Spain) at constant rate of 40 rpm. The temperature profile was set to 160 °C (feeding zone), 160 °C, 165 °C and 170 °C (die). Afterwards, the composites were pelletized to be injected using an injected molding machine Meteor 270/75 from Mateu & Solé (Barcelona, Spain) with the following temperature profile: 160 °C (feeding zone), 160 °C, 165 °C and 170 °C (die). Cavity filling time and cooling time were set to 1 s and 10 s, respectively.

### 2.4. Mechanical and Thermal Characterization

Tensile, flexural, hardness and impact testz were carried out in order to analyze the mechanical properties. Tensile and flexural testz were carried out with an Ibertest ELIB 30 universal testing machine from S.A.E. Ibertest (Madrid, Spain) at room temperature. A load cell of 5 kN and crosshead speed of 10 mm·min^−1^ were used. Sample sizes were 150 mm length, 4 mm thickness and 10 mm wide for tensile test as was indicated by ISO 527. Moreover, an axial extensometer from S.A.E Ibertest was employed to measure the tensile modulus with high accuracy. Regarding the flexural and impact tests, the samples sizes were 80 × 10 × 4 mm^3^. The impact tests were performed using a 1 J Charpy pendulum from Metrotec S.A (San Sebastián, Spain) according to ISO 179. Samples were notched with a “V” at 45° and radius of 0.25 mm. Shore D hardness mesurments were carried out with a model 673-D durometer from Instrumentos J. Bot S.A. (Barcelona, Spain) according to ISO 868. At least five samples were tested to calculate the average values and deviations.

Regarding thermo-mechanical properties, evolution in storage modulus (G’) were assessed by dynamic mechanical thermal analysis (DMTA) in an oscillatory rheometer AR G2 from TA Instruments (New Castle, DE, USA) equipped with a torsion clamp for solid samples. Rectangular sample sizing 40 × 10 × 4 mm^3^ were tested to a ramp temperature from −50 °C to 100 °C at a constant heating rate of 2 °C·min^−1^. Frequency tested was 1 Hz and maximum percentage of deformation 0.1%.

Thermal properties of samples were evaluated using Differential Scanning Calorimetry (DSC) and Thermogravimetric Analysis (TGA). DSC tests were performed in a mod. 821 DSC from Mettler Toledo Inc. (Schwerzenbach, Switzerland). Samples with a weight of 5–10 mg were evaluated using a heating program from 30 °C to 300 °C at heating rate of 10 °C·min^−1^ with a constant flow rate of 66 mL·min^−1^ in nitrogen atmosphere. The percentage crystallinity of Bio-HDPE compositions with CSF was determined by Equation (1):(1)$$X_{c}\;\left(\%\right)=\left[\frac{\Delta H_{m}-\Delta H_{cc}}{\Delta H_{m\left(100\%\right)}\cdot w_{s}}\right]\times 100$$
where $\Delta H_{cc}$ and $\Delta H_{m}$ represent the crystallization and cold enthalpies, respectively. $\Delta H_{m\left(100\%\right)}$ was the melt enthalpy of theoretically 100% crystalline of Bio-HDPE structure, which value reported in bibliography was 293 J·g^−1^ [53] and $w_{s}$ was the weight proportion of green composites.

Thermal decomposition studies were carried out in a TGA/SDTA 851 from Mettler Toledo Inc. with a heating program from 30 °C up to 700 °C at 20 °C·min^−1^ as heating rate in a nitrogen atmosphere with a constant flow rate of 66 mL·min^−1^. Samples with an average of 10 mg were employed in the TGA evaluations. Both 5% weight loss and maximum degradation temperatures were measured in order to assess the thermal stability of Bio-HDPE with different lignocellulosic filler compositions.

### 2.5. Morphology Chatacterization

The surface morphology of fractured samples in the Charpy Impact test of Bio-HDPE with CSF treated and untreated were analyzed with a Zeiss Ultra 55 Field Emission Scanning Electron Microscope (FESEM) supplied by Oxford Instruments (Oxfordshire, UK). All fractured surfaces were coated for 120 s with a thin layer of Au-Pd alloy. This process was carried out under vacuum conditions using an EM MED020 sputter coater from Leica Microsystems (Wetzlar, Germany) following the methodology employed by Quiles-Carillo et al. [54]. All samples were observed with an accelerating voltage of 2 kV.

### 2.6. Water Uptake

Water absorption was determined with samples of 80 × 10 × 4 mm^2^ size which were immersed in distilled water at 23 ± 1 °C. Previous to water immersion, samples were dried in air oven at 40 °C over 24 h to remove any residual moisture. Different samples were taken from the water, dried with a dry cloth to eliminate surface moisture and weighed for each control day (from 1 to 18 weeks). Total water absorption $(W_{abs}$) in the studied period was calculated by Equation (2):(2)$$W_{abs}\;\left(\%\right)=\frac{w-w_{0}}{w_{0}}\cdot 100$$
where w refers to the sample weight after taking it out of the water immersion bath and $w_{0}$ is the initial dry weight before immersion.

### 2.7. Colour Characterization

Colour coordinates of samples were determined in a Konica CM-3600d Colorflex-DIFF2, from Hunter Associates Laboratory, Inc (Reston, VA, USA). The colorimeter was calibrated using a white standard tile and the ASTM E313 method was used with standard illuminant D65 and observer angle of 10°. System employed was CIE Lab colour space according to the following criteria: L* represents luminance, where L* = 0 indicantes dark and L* = 100 lightness; a* represents from red (a* > 0) to green (a* < 0) and b* represents from yellow (b* > 0) to blue (b* < 0). At least five different samples were measured and the average and desviation standard values were reported. Moreover, colour difference compared to neat sample (Bio-HDPE) was calculated by Equation (3):(3)$$\Delta E_{\mathrm{ab}}^{*}=\sqrt{{\left(\Delta L^{*}\right)}^{2}+{\left(\Delta a^{*}\right)}^{2}+{\left(\Delta b^{*}\right)}^{2}}$$
where $\Delta E_{ab}^{*}$, $\Delta a^{*}$ and $\Delta b^{*}$ are the variations in colour coordinates L*, a* and b*, respectively, between neat Bio-HDPE and Bio-HDPE with CSF filler. The colour changes of samples were evaluated according to the following assesment:$(\Delta E_{ab}^{*}$ < 1) means unnoticiable colour change, $\left(\Delta E_{ab}^{*}\right)$ between 1 and 2 means a slight difference that was only noticeable by an experienced observer, $\left(\Delta E_{ab}^{*}\right)$ between 2 and 3.5 indicate that an unexperienced observer can notice the difference, $(\Delta E_{ab}^{*}$) values in the 3.5–5 range means a clear noticeable difference in colour and values higher than 5 leads to the observer noticing a different colour [55].

### 2.8. Degradation under Composting Conditions

Degadation tests were conducted under aerobic conditions following the recommendations of the ISO 20200 norm. A 300 × 200 × 100 mm^3^ synthethic compost reactor was used where the conditions of temperature and relative humidity were 58 °C and 55%, respectively. Previously to burying, all samples were dried in an air oven at 40 °C during 24 h. Six different samples of each composition were employed for each control day under composting conditions. The selected control days were: 8, 14, 21, 28, 47 and 90. Different samples of each composition were unburied while the rest remained in the process. The unburied samples were washed and dried during 24 h to remove humidity and weighed on an analytical balance. To ensure reliability, all tests were carried out in triplicate. The percetage of degradation of extracted samples was measured by Equation (4):(4)$$W\;\left(\%\right)=\frac{w_{0}-w_{s}}{w_{0}}\cdot 100$$
where $w_{0}$ referred to the initial dry weight of the sample and $w_{s}$ was the weight of the sample extracted from compost soil on different days after drying.

## 3. Results

### 3.1. CSF Extraction Yield 

Whole chia seeds were crushed in a cold press machine to extract the oil and subsequently determine the extraction yield. The extraction method was performed at room temperature for two reasons: firstly, to avoid any chemical changes in the extracted oil that can be induced by high temperatures and to use less energy in the extraction, thus reducing operation costs. Moreover, mechanical extraction is preferable to a chemical extraction because it avoids the use of petrochemical solvents. After a second extraction process a yield of 67.7 wt.% was obtained. No more extraction trials were carried out due to very low yield obtained, considered non-economically feasible as Ixtaina et al. reported [56]. Therefore, this elevated amount of residual cake (67.7%) produced after extraction of CO, was considered a potential candidate to apply in green composites as renewable fillers contributing to the circular economy.

### 3.2. First Stage. Effect of Silane Treatment

The effect of untreated and treated CSF filler in a Bio-HDPE matrix was investigated by evaluating the tensile, ductile, hardness and impact properties. Table 2 gathers the properties obtained. Firstly, the tensile strength of Bio-HDPE was around 373 MPa and decreased with the addition of CSF filler (untreated and treated). This behavior was expected due to the addition of filler, which causes a decrease in interfacial adhesion because the filler tends to form agglomerates increasing the stress concentrations and, as a result, lower tensile strength [57]. Regarding the effect of silane treatment, which was evaluated with 20 wt.% of CSF, it was observed that the 20UT sample presented 10.8% less tensile strength than the 20T sample. This reduction of tensile strength was attributed to weak interfacial adhesion between UTCSF and the Bio-HDPE matrix. It is known that organosilanes consist of two different reactive molecules: the silanol groups react with the hydroxyl groups presents in CSF, whereas the functional groups of the APS react with the polymeric matrix by covalent bonding [58]. In addition, it has been reported that treatment of lignocellulosic flour with APS coupling agent improved the interfacial adhesion with Bio-HDPE [59]. This better adhesion/interaction allowed a uniform stress distribution from polymer to filler, achieving a higher tensile strength than untreated samples. Similar findings have been reported by Ihamounchen et al. [60], who evaluated the addition of olive husk flour (OHF) treated with vinyltriacetoxysilane (OHFTA) in HDPE. In this study an increment from 13% to 3% in tensile strength with 10 and 30 wt.% of OHF was reached compared to untreated samples, respectively. Regarding tensile modulus, the untreated sample (20UT) presented a value of 374 MPa whereas a treated sample (20T) showed an increment of 5.9% compared to the 20UT sample. Finally, elongation at break was dramatically decreased after filler addition. Bio-HDPE presented a high elongation at break of 520%, showing high ductility properties as has been reported previously [44]. When 20 wt.% of untreated CSF was added, this parameter dropped drastically to a value below 36%, which is 93% lower than Bio-HDPE. On the other hand, if this amount of filler is treated with APS the decrease is pronounced, but lower than 90% with respect to the Bio-HDPE. Therefore, it has been observed that CSF treated with APS presented higher elongation at break than a 20UT sample with an improvement of 41%. Comparing these results with the use of a polypropylene-graft-maleic copolymer (PP-g-ma) as compatibilizer between peanut shell and polyethylene, lower enhancement in elongation at break were obtained [61]. In this study the introduction of PP-g-ma does not improve this feature compared to sample without compatibilizer, which highlights the efficiency of the APS.

With regard to flexural properties, no significant changes were recorded in flexural strength with the addition of CSF filler. In addition, the silane treatment with APS does not show significant differences. Regarding flexural modulus of the 20UT sample, a decrease from 804 MPa for neat Bio-HDPE to 784 MPa was observed. Regarding samples with CSF treated with APS, an evident improvement was observed compared to 20UT samples. The same behavior was obtained by Boronat et al. who evaluated Bio-HDPE with eggshells with silane treatment [44]. In another study, a maleic anhydride grafted polystyrene (Xibond^TM^ 160) was employed as a compatibilizer in a cellulose/ABS composite [62]. The addition of this copolymer led to a 1.75% improvement of the flexural modulus compared to a sample without compatibilizer. This value was lower than that obtained with APS in the current study that achieved an improvement of 7% regarding the untreated sample. In general, these tendencies were in concordance with values previously recorded for tensile properties, where samples treated with silane present higher mechanical resistances properties than untreated samples.

Regarding Shore D hardness, it increased with addition of CSF due to the intrinsic hardness of lignocellulosic filler which leads to an increase of the hardness of the composite [63]. In relation to untreated and treated CSF, there was a slight improvement with the 20T sample. This fact was attributed to the strong adhesion between CSF and the matrix produced after silane treatment [64]. On the other hand, the impact energy, that is related to the deformation capacity, is highly sensitive of stress concentrators [65]. The addition of 20 wt.% of CSF led to a decrease in impact absorbed energy compared to Bio-HDPE. In this instance, it has not been possible to transfer the impact load from the matrix to the filler thus reducing the impact absorbed energy with CSF addition. Comparing 20UT and 20T samples, an improvement of 13% in impact energy was observed when filler was treated with APS. This behavior was due to the silane coupling agent acting as a bridge between CSF and Bio-HDPE matrix through chemical bonding, which reduces any crack propagation by means of a good dispersal of the impact energy [66].

In Figure 1, the morphology of fractured surfaces from Charpy tests are shown with the aim of evaluating the effect of silane treatment on the interfacial adhesion between CSF and the matrix. As can be observed in Figure 1a, Bio-HDPE showed the typical rough and irregular surface of a ductile polymer as also reported by Rojas et al. [67]. As it has been described previously, the treatment of CSF with APS has a positive effect on mechanical resistance properties such as strength, modulus, or hardness whereas a decrease of mechanical ductile properties such as elongation at break was detected. The effect of silane treatment can be compared in Figure 1b,c at 1000× where CSF particles are highlighted with a yellow arrow. In the case of an untreated sample (20UT) a clear gap can be distinguished in the perimeter between the polymer matrix and a CSF particle highlighted by the red arrow in Figure 1b. This gap indicated the lack of particle-matrix interaction that does not allow it to transfer stress, and this justifies the decrease in elongation at break and impact energy [60]. Moreover, Toro et al. indicated that when cracks are produced by an impact, these are propagated towards the poor interfacial regions, leading a break in the composites with low stresses [68]. Regarding the treated sample (20T), the gap between the CSF particle and polymer matrix has been broadly reduced when CSF was treated with silane coupling, as has been highlighted with a red arrow in Figure 1c. This indicated that a better interfacial adhesion was achieved between the two phases. This confirms the enhancement of mechanical properties and elongation at break compared to untreated samples. Bijaisoradat et al. also observed that voids between filler and matrix was reduced when evaluating wood flour treated by trimethoxy (propyl)silane (MPS) in HDPE [69].

The effect of untreated and treated CSF filler in the Bio-HDPE matrix was studied by DMTA in torsion mode. Figure 2 shows the storage modulus (G’) of samples with respect to the temperature. Firstly, Bio-HDPE presented the lowest G’ as was expected. As has been mentioned previously, the tensile and flexural modulus presented a clear increase with 20UT and 20T samples compared to Bio-HDPE. In this case, the same trend was observed because of the restriction of polymer chain mobility through addition of CSF filler, hence increasing the stiffness of the composite [70]. Regarding the effect of silane treatment, it was observed that 20T sample presented a higher G’ than the 20UT sample over all temperatures, which means an increase of 2.5% at room temperature. This fact csn be explained by the better compatibility between CSF and matrix as was observed in FESEM. These values were in concordance with the mechanical properties discussed above.

The influence of silane coupling treatment on the thermal properties has been investigated by DSC. Table 3 gathers the main thermal properties. Bio-HDPE showed a melt temperature of 131 °C, a value in concordance with the result of Quiles-Carillo et al. [71]. In addition, it was observed that the melting temperatured of 20UT and 20T samples are not affected significantly compared to Bio-HDPE. It has been widely reported that silane coupling agent does not directly affect the melting temperature [59,72,73]. With respect to the melting enthalpy, it was decreased with addition of CSF filler due to both the effect of filler content and the matrix reduction (Bio-HDPE). Finally, the effect of silane treatment on crystallinity is shown in Table 3. The addition of treated CSF (20T) allowed an increase in the crystallinity compared to untreated CSF (20UT). This behavior was due to a better interaction between phases that may improve the nucleation activity with the presence of CSF treated with APS [74]. Therefore, the lack of interaction of untreated CSF led to difficult the arrangement of molecular chains, decreasing the crystallinity.

Thermogravimetric analysis (TGA) was performed to assesses the thermal stability of neat Bio-HDPE, CSF particles and the effect of silane treatment with 20 wt.% of CSF in the Bio-HDPE matrix. Figure 3 shows the TGA curves and the first derivative (DTG). As observed in Figure 3a, a weight loss of Bio-HDPE was associated with a single-phase degradation. At a temperature of 100 °C no weight loss due to evaporation of residual moisture was measured in the samples, due to their characteristic hydrophobic nature. Bio-HDPE degradation proceeded in one step that began about 350 °C up to 520 °C where a weight loss of 99% was measured. This one step degradation was produced by random scission of the C-C and C-H bonds present in Bio-PE [75]. A similar temperature profile trend was observed by Montanes et al. [76]. Regarding the weight loss of CSF, it was associated with the four stages characteristic of lignocellulosic particles. The initial weight loss recorded was 7.5% in the 30–220 °C range, which was attributed to evaporation of residual moisture contained in the particles [77]. The second step was produced between 220 °C and 350 °C approximately, with a weight loss of 40%. This loss was due to the decomposition of low molecular weight products such as hemicellulose. The third stage corresponds to around 60% of weight loss in the 350–410 °C range due to decomposition of cellulose. The last step, characterized by a weight loss of 75%, was related to lignin degradation at temperatures above than 410 °C [78]. Two different stages were identified in the evaluation of the effect of CSF particles in the Bio-HDPE matrix. The first stage, at temperatures in the 280–430 °C range, was related to the lignocellulosic filler decomposition mentioned above. It was observed that addition of CSF filler led to decrease in the thermal stability. The second stage was measured above 430 °C and is characterized by Bio-HDPE degradation. Regarding the effect of silane treatment, a treated sample (20T) displayed higher thermal stability than an untreated (20UT) one. This could be attributed to the improvement of matrix filler interactions after silane coupling agent addition, which led to enhanced thermal stability [64]. In Figure 3b two maximum degradation stages can be observed for all samples. The first stage was related to CSF degradation at temperatures around 350 °C, thus not displaying any evidence of weight loss for Bio-HDPE as expected. In addition, the maximum temperature for the first stage was not shifted depending on the CSF filler surface treatment. Finally, the second stage was related to Bio-HDPE degradation. In this case, the second maximum degradation temperature was 498 °C and no signs of changes were recorded, maintaining the thermal stability. However, the results reported by Fonsenca et al. showed a shift to lower temperatures of the peak maximum peak degradation of ABS with a cellulose fiber composite using a maleic anhydride grafted polystyrene as compatibilizer, reducing the thermal stability [62].

With the objective of measuring the effect of silane treatment on the water uptake of the developed composites, the absorption of water was evaluated by means of immersion in distiller water for 18 weeks. In Figure 4 it can be observed that Bio-HDPE presented less than 0.05 wt.% water absorption. It is known that polyethylene is a hydrophobic polymer as has been reported by Perthue et al. [79], who reported the same absorption value. In addition, CSF presents the hydrophilic nature characteristic of lignocellulosic fillers. Thus, the addition of filler leads to an increase in the water absorption of composites. The free hydroxyl groups present in CSF may react with hydrogen bonds of water allowing water to diffuse inside the composite [80]. The effect of silane treatment on CSF was clearly observed by comparing 20UT and 20T samples. A decrease of 17% in water absorption was recorded for the 20T sample compared to the 20UT sample. After silane treatment there is less availability of hydroxyl groups in CSF filler to create hydrogen bonding with water, thus rendering lignocellulosic filler more hydrophobic [81]. In addition, we should remark that from 14 to 18 days, a constant water uptake was measured for all samples studied, indicating no further absorption.

Degradation of Bio-HDPE with untreated and treated CSF filler is shown in Figure 5. Bio-HDPE presents several features such as high molecular weight, hydrophobicity and the lack of functional groups that are recognizable by microbial systems, which make it essentially a non-biodegradable polymer [82,83]. After 8 days, Bio-HDPE does not show any weight loss whereas 20UT and 20T samples presented a slight weight loss. On the 21st day a weight loss higher than 2.5% was recorded for both untreated and treated samples with 20 wt.% of CSF. According to Peng et al. [77], lignocellulosic filler is biodegradable at a reasonable rate and can be fully degraded after longer periods ranging from 1 to 3 months using the soil burial method. The recorded degradation was related to the lignocellulosic filler which biodegrades due to the deterioration of lignin, hemicellulose and cellulose caused by microorganisms [84]. It should be mentioned that the Bio-HDPE matrix was not affected after 90 days, displaying a weight loss of less than 0.05% as plotted in Figure 5. Regarding to effect of silane treatment, the 20UT sample achieved a higher disintegration rate than 20T sample, reaching 8.5% and 4.21% after 18 weeks, respectively. It is known that hydrophilic nature of lignocellulosic filler facilitates the transfer of water and microorganisms or enzymes into the composite, thus speeding up the disintegration process [74]. The improvement of interfacial adhesion due to silane treatment hinders the introduction of water and microorganism action in composites, reducing the degradation [74]. This behavior was in concordance with the water uptake results obtained, where the 20UT sample present a higher water absorption than a 20T sample, thus leading to a sped-up disintegration rate.

After studying the effect of silane coupling agent in Bio-HDPE with CSF filler, it has been determined that in general, silane treatment improves the mechanical resistance properties and main aspects such as the water uptake capacity and minimizes the disintegration rate under composting conditions. By FESEM it has been observed how the gap between filler and matrix was reduced with silane treatment, which justifies the mechanical property improvement. Moreover, silane agent provided a nucleation effect, a decrease of the water absorption due to decrease of available hydroxyl groups and thus slightly decreased the rate of degradation.

### 3.3. Second Stage. Evaluation of CSF Filler Percentage

Once the effect of APS as a coupling agent had been studied, the aim was to study how CSF filler content modified with APS affects the mechanical, thermal, and morphological properties of the Bio-HDPE. The same order of characterization as in the first stage was followed.

Tensile properties of Bio-HDPE with different content of CSF are shown in Figure 6. As can be seen in Figure 6a, the tensile strength decreased with increasing filler content. The lowest tensile strength recorded was 8 MPa for the 40T sample, which was a decrease of 58% compared with Bio-HDPE. This behaviour was directly related to the interfacial adhesion between CSF and matrix, which is crucial in reinforced composites to allow the tranfer of a small stress to filler particles during deformation [45]. Therefore, a higher content of filler implies less interaction between the matrix and particles, and thus more stress concentration appeared that induces the strength to decrease [85]. Regarding the tensile modulus, Figure 6b showed an increase with addition of CSF content in the matrix. In this case, the highest elastic modulus was obtained for the 40T sample, which represents an increment of 22% compared to neat Bio-HDPE. No significant difference was observed when comparing the 30T and 40T samples, indicating that a higher addition of CSF filler does not necessarily lead to an increase of elastic modulus. In addition, the increment of elastic modulus could confirm that CSF presents the rigid nature typical of lignocellulosic fillers. This trend indicated an increment of stiffness of composite due to restriction of chain mobility caused by the presence of filler particles [63]. Finally, as was shown in Figure 6c, the elongation at break of Bio-HDPE decreased dramatically as the filler content increased, going from 520% for Bio-HDPE to 23.5% for the 40T sample, respectively. This behaviour was attributed to the presence of stress caused by the dispersed rigid filler that increased the higher the filler content is [61]. This trend has also been found by Chun et al. when studying the effect of coconut shell powder (CSP) with 3-aminopropyl-triethoxysilane (3-APE) silane coupling in a PLA matrix [72]. In this study, despite the enhancement of interfacial adhesion obtained using aminosilane, a reduction of elongation at break occurred.

A similar behaviour was observed in flexural properties (Figure 7). As shown in Figure 7, the flexural strength decreased slightly with the addition of CSF to the matrix. Although the changes recorded are not very significant, a clear tendency was observed. The lowest flexural strength measured was 21.7 MPa for 40 wt.% of CSF, representing a decrease of 9.6% compared to Bio-HDPE with 24 MPa. With respect to flexural modulus, it was highly improved with the addition of filler. Addition up to 20 wt.% of CSF showed an increase of 4% compared to unfilled material, achieving an average value of 845 MPa approximately. However, a very similar addition of 30 wt.% showed a significant increment of 10% and 18% in flexural modulus regarding Bio-HDPE for 30 and 40 wt.%, respectively. As mentioned above, for the lowest filler content (10 and 20 wt.%) a slight increase was measured. This fact is due to better filler dispersion in the matrix as well as the substantial distance between filler particles. Nevertheless, the flexural modulus for high filler content was increased noticeably because the distance was decreased by the addition of filler and the effect of each CSF particle was superposed providing an evident enhancement [86]. In general, these tendencies were in concordance with the previously recorded tensile properties, where the tensile modulus increased, and tensile strength decreased as filler content was added.

Shore D hardness and impact absorbed energy results are gathered in Table 4. Firstly, hardness is related to the mechanical resistance properties and a gradual increase was observed as the addition of CSF filler increased. In this case, the 40T sample showed a Shore D hardness of 63, which represents an increment of 11% compared to Bio-HDPE. This fact was expected because lignocellulosic filler presents higher hardness than the soft polymer matrix, which leads to an increase in the hardness of composites [87]. It should be noted that the evolution of hardness was in concordance with the tensile and flexural modulus results. In contrast, the toughness of composites, which is the ability to absorb energy during deformation, was decreased as the addition of CSF to the matrix increased. The lowest impact absorbed energy recorded was 1.6 kJ·m^−2^ for 40 wt.% of CSF, which represents a decrease of 38% compared to Bio-HDPE (2.6 kJ·m^−2^). As expected, filler addition generated higher stress concentration as well as restricted the deformation, which leads to decreased absorbed energy [88]. Furthermore, this tendency was in concordance with the evolution of elongation at break and the typical mechanical ductile properties mentioned above.

In Figure 8 the fractured surface morphology after impact tests of Bio-HDPE with different percentages of treated CSF with APS are shown in order to evaluate filler-matrix interactions. In Figure 8a the typical irregular and rough surface of a ductile polymer like Bio-HDPE is shown, where an an absence of CSF particles is observed. In general, after addition of CSF particles, highlighted in yellow arrows (Figure 8b–e), good bonding by Bio-HDPE matrix was observed. As mentioned previously, CSF particles treated with APS showed improved compatibility with matrix reducing the gap between phases. Nevertheless, as higher CSF particle addition levels, more presence of small gaps in the perimeter between CSF and the surrounding Bio-HDPE matrix was observed as marked by orange arrows. It is known that a higher presence of particles in the matrix, despite using compatibilizers, generates more voids between particles and the matrix [89]. This fact causes an increment of stress concentrators and therefore a loss of mechanical properties as was shown previously [85]. Similar findings were obtained by Garcia-Garcia et al. who evaluated different percentages of peanut shell powder (PSN) in a Bio-HDPE matrix, where despite the use of compatibilizers, some voids between particles and the matrix causing a decrease in elongation at break and tensile strength are still observed [61].

DMTA in torsion mode, plotted in Figure 9, represents the evolution of storage modulus (G’) with respect to temperature for Bio-HDPE and different composites developed with Bio-HDPE and CSF. Storage modulus, which is related to the stiffness of a material, decreased as the temperature increased for all analyzed samples. However, the addition of filler increased the storage modulus values regarding neat Bio-HDPE, with higher values as the CSF content increases. This was more clearly observed at low temperatures (−100 to −80 °C). With respect to room temperature, it was observed that the 40T sample presents a G’ of 105 MPa, which represents an increment of 20% compared to neat Bio-HDPE. This fact confirmed the reinforcing effect that CSF filler provides, which acts as an interlock point in Bio-HDPE matrix that leads to restricted chain mobility, thus increasing the stiffening behaviour of composites [63]. This trend was also measured previously in tensile and flexural modulus. In addition, a similar behavior has been reported by Barczewski et al. [20] who evaluated different linseed cake (LC) percentages in HDPE composites. In this study, an increase of G’ with addition of LC was also observed, being more noticeable at low temperatures.

Thermal analysis was performed in order to assess the main thermal transitions and thermal stability for Bio-HDPE and different composites developed with Bio-HDPE and CSF. Thermal parameters obtained by DSC are gathered in Table 5. It was observed that the addition of CSF to the Bio-HDPE matrix barely affects the melting temperature. Regarding crystallinity, which is directly related to melting enthalpy ($\Delta H_{m}$), it was increased with the presence of a low amount of filler (10 and 20 wt.%). This increment in the crystallinity was due to the nucleating effect provided by the CSF filler [90]. However, a CSF content higher than 20 wt.% led to a decrease of the crystallinity with the lowest value being 49.1% for 40 wt.% of CSF. In general, the crystallinity of composites reinforced with particles depends on two factors: the first is the nucleation effect of fillers and the second is the hindering effect produced by fillers that difficult the movement of molecular chains. However, although filler presented a nucleation effect, high amounts of filler such as 30 and 40 wt.% lead to difficulties in the arrangement of molecular chains, thus decreasing the crystallinity [91]. This tendency was also reported by Xiong et al. who evaluated HDPE with wood flour with different coupling agents [59].

In Figure 10 both TGA curves and their first derivative (DTG) of different composites developed with Bio-HDPE and CSF are plotted. Bio-HDPE presented a temperature for a weight loss of 5 wt.% (T_5%_) of around 460.3 °C. The addition of CSF particles in the Bio-HDPE matrix led to a decrease of T_5%_ as CSF filler was added. As shown in Figure 10a, T_5%_ was reduced from 460.3 °C (Bio-HDPE) to 262.3 °C for the 40T sample, which means a decrease of 43%; hence, reduced thermal stability compared to neat Bio-HDPE. This decrease of T_5%_ is due to the decomposition of the lignocellulosic filler produced in the 220–410 °C range, as exposed previously. In addition, a higher temperature of 410 °C was related to Bio-HDPE degradation, not showing significant changes. Finally, the residual weight recorded from 520 °C to 700 °C, was higher as CSF filler was added, probably due to the presence of more ash content [92]. In Figure 10b the first derivative is shown, where two maximum degradation stages in composite reinforced with CSF are clearly observed. The first stage (T_max1_) was related to CSF filler degradation which occurred at about 338 °C, and the second stage (T_max2_) with Bio-HDPE degradation at 498 °C. As can be observed, the addition of CSF to the matrix does not affect the T_max1_ and T_max2_. Similar findings have been reported by Yong et al. [93] for wood plastic composites using polyethylene and wood fiber with different formulations.

Water absorption of Bio-HDPE with different percentages of CSF filler have been measured. The results obtained after 18 weeks are shown in Figure 11. This phenomenon depends on the capabilities of matrix and filler to absorb water. Firstly, Bio-HDPE presented less than 0.05% of water absorption which confirms its hydrophobic nature. The addition of CSF filler provided an increment of water absorption due to the hydrophilic nature of CSF, as was described previously. In agreement with the literature, initially a higher water absorption was observed that gradually slows down until saturation was achieved, being in this case at 18 weeks [94]. The addition up to 20 wt.% presented a slight increase in water absorption reaching almost 3%. However, a notable increase was observed for 30 and 40 wt.% of CSF, where values of 8.25% and 11% were obtained, respectively. This increment of water absorption could be attributed to the presence of mucilage in CSF, which is a polysaccharide gum that represents about 6% of the weight and provides higher water-holding capacity [80]. Chen et al. [95] studied the water uptake of rice husk (RH) in a HDPE matrix, reaching values of about 7% with 40 wt.%. Furthermore, Liu et al. [96] has also reported a water absorption of less than 11% with 50 wt.% of wood flour in a PP matrix, observing a lower value compared to a 40T sample. Nevertheless, general values of water absorption in WPCs are around 14–16% [95,97,98]. Therefore, although Bio-HDPE with CSF particles presented higher values than some WPCs, probably due to the presence of mucilage, it remained below that of general WPCs.

The appearence of materials with fillers are crucial to imitate wood as closely as possible. Parameters such as colour or luminance are important in this approach. Colour coordinates of different Bio-HDPE/CSF composites are gathered in Table 6 by the values of L*(luminance), a* (green to red), b* (blue to yellow) and colour change (ΔE_ab_^*^). Moreover, the visual appearance of samples is shown in Figure 12. Firstly, the sample with highest luminance was Bio-HDPE, caused by its white colour with a value of 71.3. With the addition of lignocellulosic filler, luminance was reduced in the range of 36.6–42.5 compared to neat Bio-HDPE. Regarding a*, Bio-HDPE presented a negative value of −2.6 which was close to 0 that indicates the closeness to white colour. On the contrary, the rest of samples had positive values due to the characteristic brown colour of chia seed flour [99]. Higher values of a* coordinate have been reported by Jorda-Reolid et al. [100] who employed Bio-HDPE with argan shell wastes. In this study a higher a* value above 5 was reported, thus a reddish brown colour was obtained [100]. Referring to colour coordinate b*, it is an indicative of blue to yellow colours. Bio-HDPE presented a negative value of −2.68 which is in concordance with Rojas et al. [67]. The rest of samples had values between 5.72 and 8 which indicate a tonality approaching yellow. Regarding the colour change variation, it should be noted that a difference of colour change between Bio-HDPE and CSF composites exists. This change was more noticeable as the addition of CSF fillers was increased.

The disintegration process of Bio-HDPE with different percentages of CSF is plotted in Figure 13. As it was mentioned previously, Bio-HDPE is a non-biodegradable polymer and no sign of weight loss was recorded after 90 days. The absence of hydrolyzable zones in its structure that can be attacked by microorganisms means that after 90 days of testing, the disintegrated mass is almost nil. However, the addition of CSF filler, as expected, increased the disintegration rate up to 6% weight loss after only 8 days for a 40T sample. It is known that lignocellulosic filler, which is composed of lignin, hemicellulose and cellulose, is biodegraded by microorganisms [84]. Disintegration weight loss was greatly increased up to 21 days with addition of filler, where 16.5% of weight loss was measured in the 40T sample. However, from 21 to 90 days only a slight increase from 16.5% up to 20% was shown by the 40T sample. The applied standard, ISO 20200, establishes that any substance from developed composites that is able to pass through a 2 mm sieve is considered as a degraded material. These disintegrated materials are composed mostly of lignocellulosic fillers and possibly microplastics. The eventual microplastic formed may be further degraded by other additional mechanism such as thermo-oxidative or photo-oxidative degradation, leading to formation of a more hydrophilic layer then suitable for microorganism degradation [101]. This last mechanism will take a longer time as well as will depend on abiotic factor before microorganisms can assimilate it. Therefore, following the definition of biodegradable polymer established by the UNE EN 13432 standard, which establishes that the disintegration rate after 90 days of testing must be greater than 90% by weight, the composite developed cannot be considered as totally biodegradable, but it can be considered partially biodegradable. Another crucial aspect was the visual appearance of composites after 12 weeks, which is shown in Figure 10. With the addition of CSF, the appearance change was more noticeable due to the disintegration rate compared to the initial day (Figure 14), and 30T and 40T samples displayed a more brittle aspect. Therefore, it could be concluded that, although is not considerable a fully biodegradable WPC, it is partially biodegradable.

Therefore, although a fully biodegradable WPC is not obtained, the addition of a residue from the extraction of chia oil allows the development of a composite with the balanced properties provided by the bio-HDPE and with a biodegradability of around 20% by weight of the composite after 12 weeks of testing. Among interesting areas of applications, the packaging sector could be highlighted due to its partially biodegradability in order to reduce the environmental impact.

## 4. Conclusions

In this work, a new composite was obtained from the residual waste remaining after chia seed oil extraction. In the first stage, the effect of a silane coupling agent [3-(2-aminoethylamine) propyl]-trimethoxysilane (APS) treatment were examined by comparing untreated and treated samples with 20 wt.%. The improvement of interfacial adhesion between treated CSF with APS and Bio-HDPE matrix was observed by FESEM. As a result of this better interaction, an enhancement in general mechanical properties was obtained compared to untreated CSF, with an improvement of 41% in elongation at break. Regarding water uptake, the sample with APS recorded a decrease of 17% compared to untreated sample, caused by the reduction of available hydroxyl groups in the lignocellulosic filler. Consequently, in the test of disintegration under composting conditions, a sample treated with APS recorded a reduction of the weight loss of 50% due to its more hydrophobic nature resulting from the silane treatment which hinders the penetration of water and microorganisms.

In the second stage, an optimization of the amount of CSF filler has been carried out. The addition of 40 wt.% of CSF resulted in improvements of 22% and 18% in tensile and flexural modulus compared to neat Bio-HDPE, respectively. The addition of CSF caused a slight decrease of thermal stability due to the presence of lignocellulosic filler, but aesthetically, the characteristic brown colour of the composites developed made them suitable to for manufacturing furniture or food packaging. The degree of water uptake was similar to that of other wood plastic composites on the market, while the study of disintegration by composting shows a weight loss ratio 20% higher than that of Bio-HDPE, a positive aspect to be taken into account in the development of new composites for sectors with products with short life cycles.

## Figures and Tables

**Figure 1 polymers-13-02269-f001:**
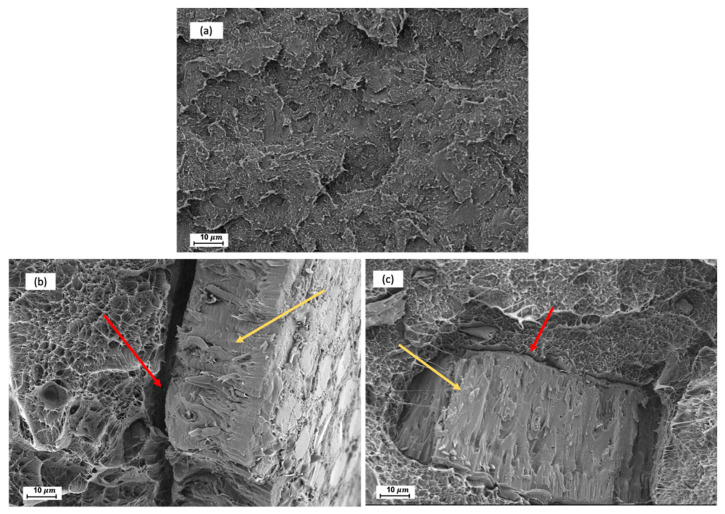
Fracture surface morphology of Charpy test by FESEM at 1000x: (**a**) Bio-based high-density polyethylene; (**b**) 20 wt.% untreated chia seed flour; (**c**) 20 wt.% treated chia seed flour.

**Figure 2 polymers-13-02269-f002:**
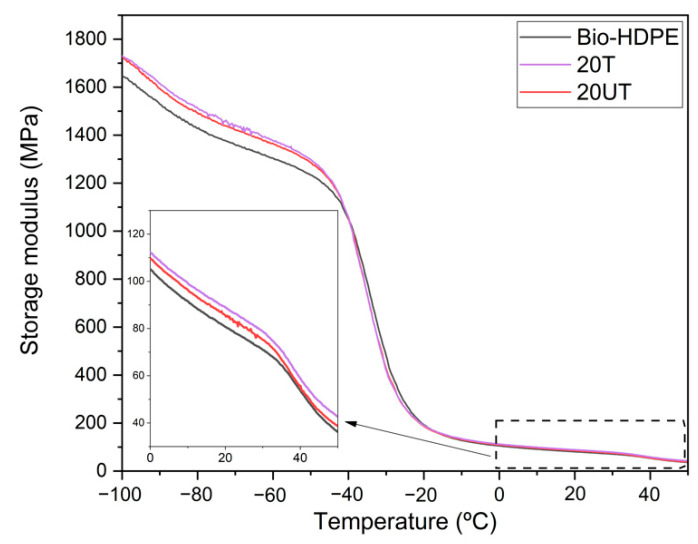
Evolution of storage modulus (G’) of bio-based high-density polyethylene (Bio-HDPE) with 20 wt.% of untreated (20UT) and treated (20T) chia seed flour.

**Figure 3 polymers-13-02269-f003:**
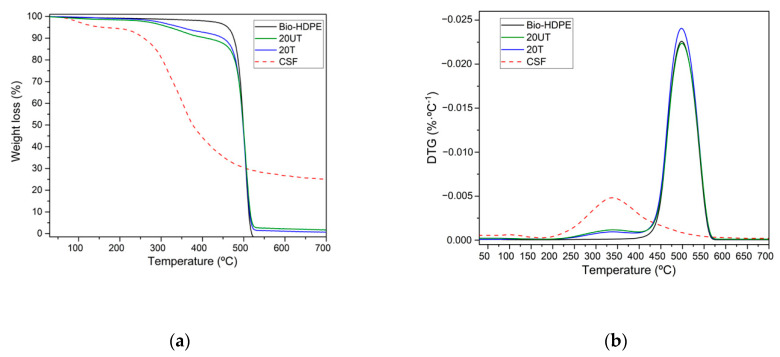
Thermal parameters of degradation of bio-based high-density polyethylene (Bio-HDPE) with 20 wt.% of untreated (20UT) and treated (20T) chia seed flour (CSF). (**a**) Weight loss; (**b**) Derivative thermogravimetry.

**Figure 4 polymers-13-02269-f004:**
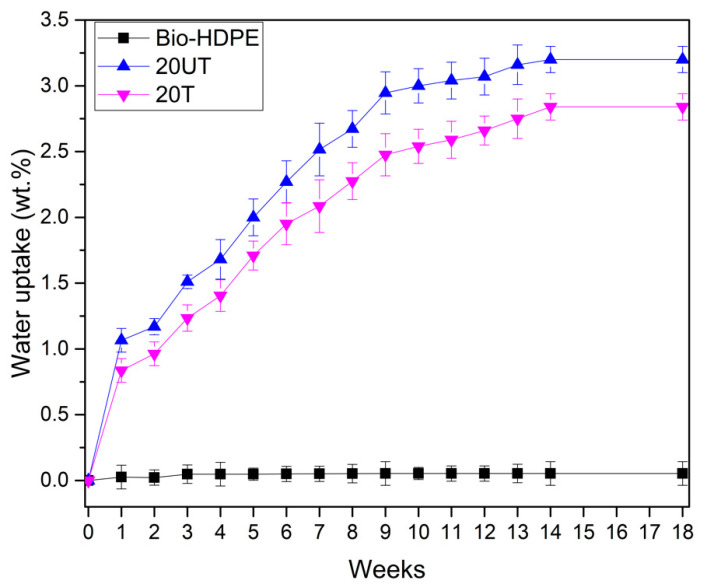
Water uptake of bio-based high-density polyethylene (Bio-HDPE) with 20 wt.% of untreated (20UT) and treated (20T) chia seed flour.

**Figure 5 polymers-13-02269-f005:**
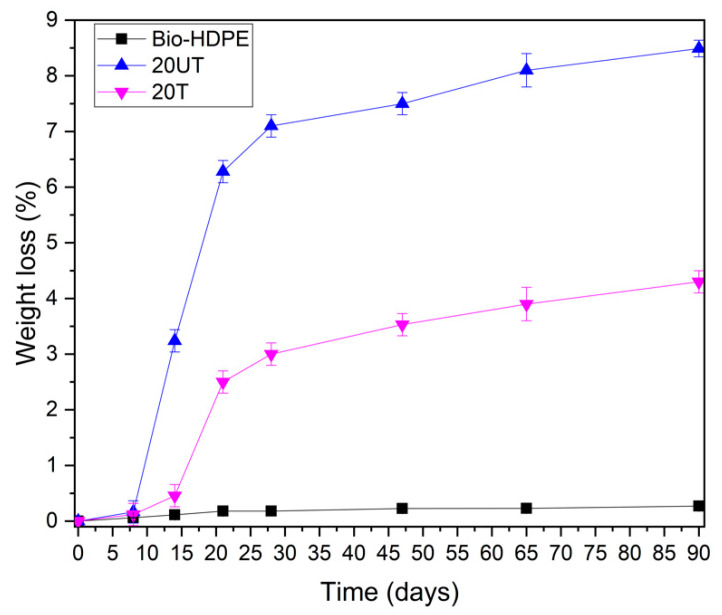
Weight loss of bio-based high-density polyethylene (Bio-HDPE) with 20 wt.% of untreated (20UT) and treated (20T) chia seed flour.

**Figure 6 polymers-13-02269-f006:**
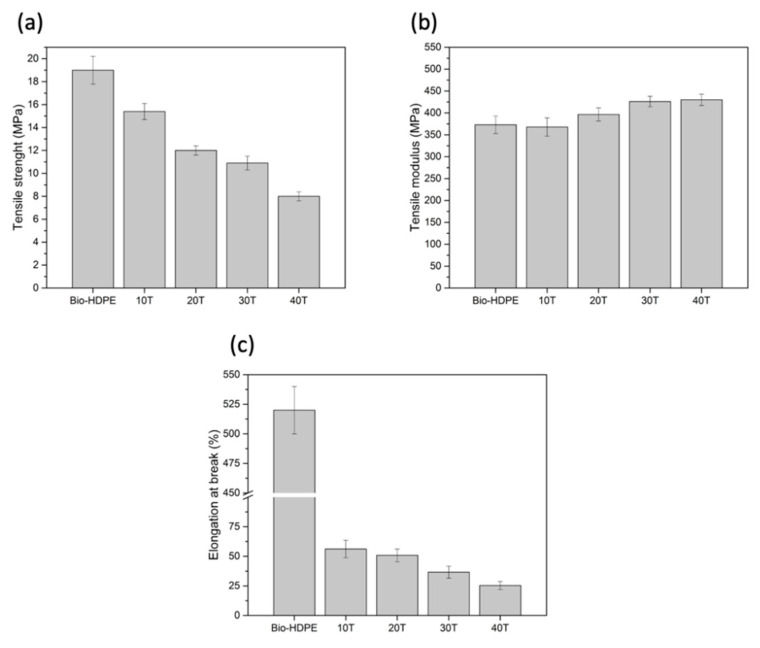
Effect of weight percentage of treated chia seed flour with APS on tensile properties in bio-based high-density polyethylene (Bio-HDPE): (**a**) Tensile strength; (**b**) Tensile modulus; (**c**) Elongation at break.

**Figure 7 polymers-13-02269-f007:**
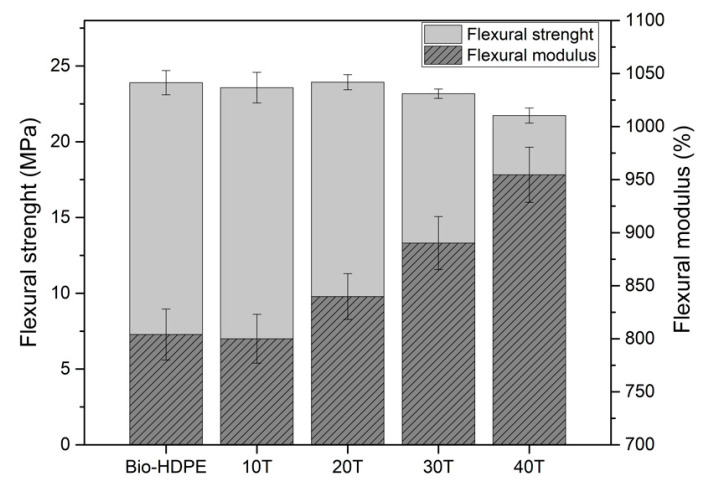
Effect of weight percentage of treated chia seed flour with APS on flexural properties in bio-based high-density polyethylene (Bio-HDPE).

**Figure 8 polymers-13-02269-f008:**
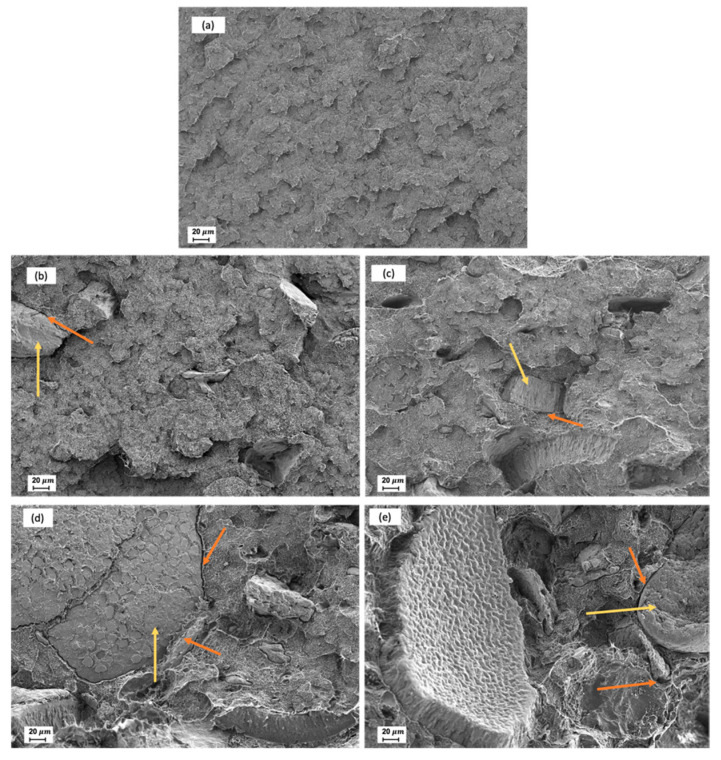
Fracture surface morphology of Charpy test by FESEM at 250 × with different percentage of treated chia seed flour: (**a**) Bio-based high-density polyethylene; (**b**) 10 wt.%; (c) 20 wt.% (**d**) 30 wt.%; (**e**) 40 wt.%.

**Figure 9 polymers-13-02269-f009:**
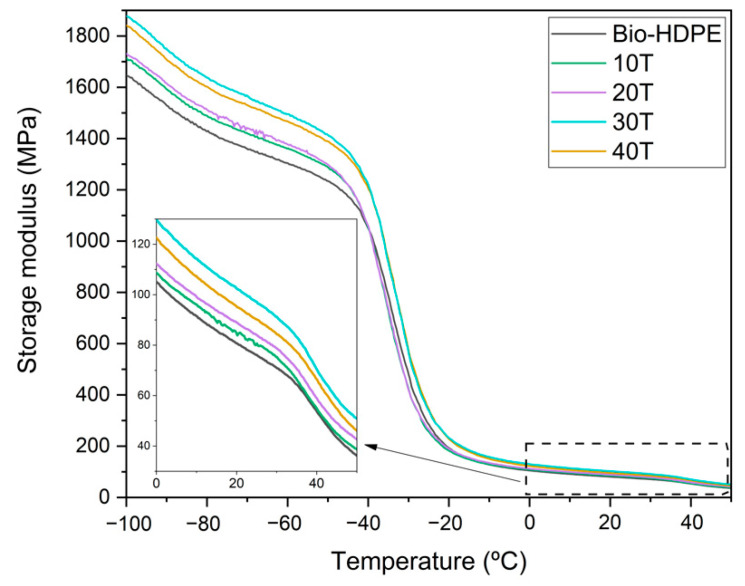
Evolution of storage modulus (G’) in bio-based high-density polyethylene (Bio-HDPE) with different weight percentage of treated chia seed flour with APS.

**Figure 10 polymers-13-02269-f010:**
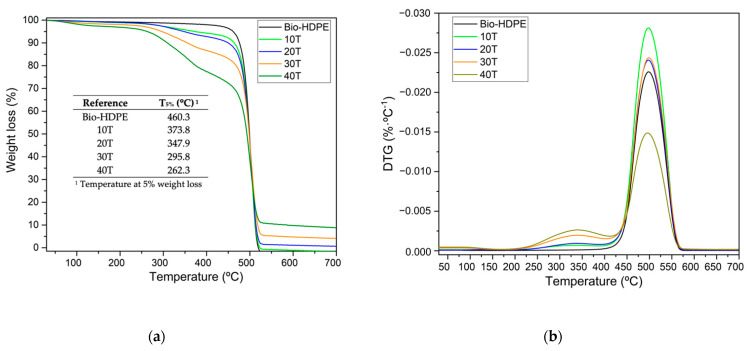
Thermal parameters of degradation of bio-based high-density polyethylene (Bio-HDPE) with different weight percentage of treated chia seed flour with APS. (**a**) Weight loss; (**b**) Derivative thermogravimetry. T_5%_ is temperature at 5% weight loss; T_max1_ and T_max2_ is maximum degradation temperature for first and second stage.

**Figure 11 polymers-13-02269-f011:**
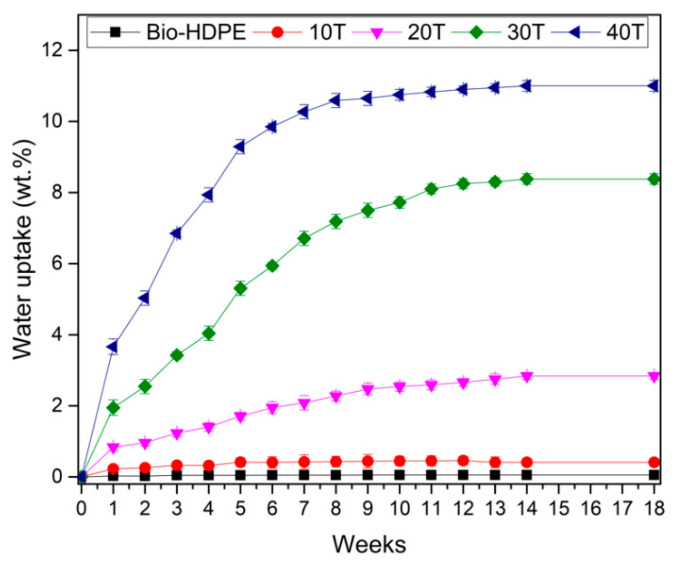
Water uptake of bio-based high-density polyethylene (Bio-HDPE) with different weight percentage of treated chia seed flour with APS.

**Figure 12 polymers-13-02269-f012:**
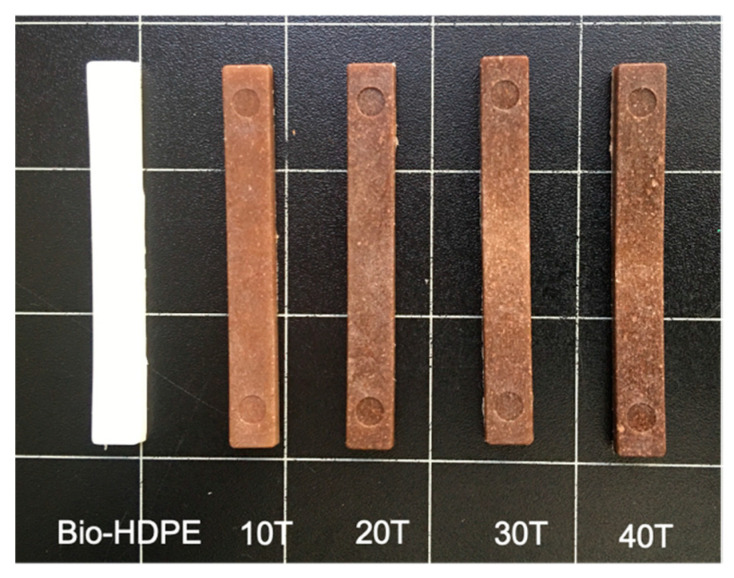
Visual appearance of bio-based high-density polyethylene (Bio-HDPE) with different weight percentage of treated chia seed flour with APS.

**Figure 13 polymers-13-02269-f013:**
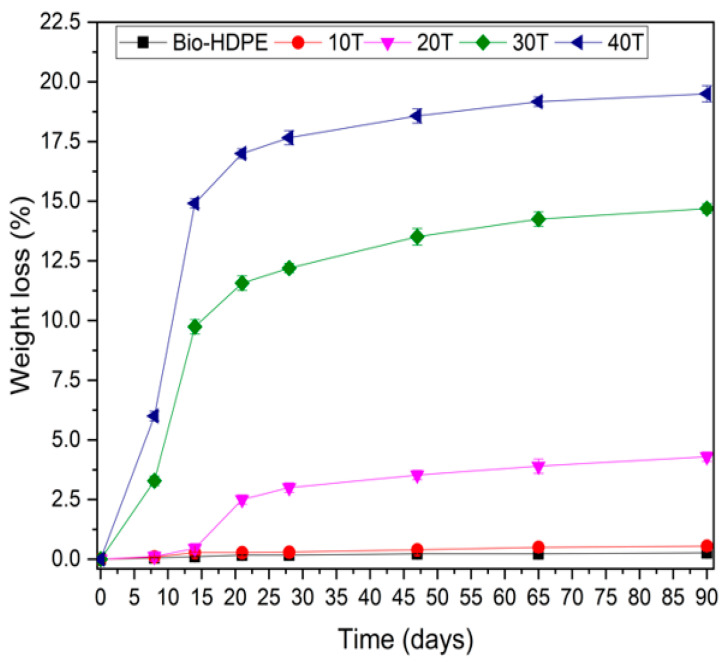
Weight loss of bio-based high-density polyethylene (Bio-HDPE) with different weight percentage of treated chia seed flour with APS.

**Figure 14 polymers-13-02269-f014:**
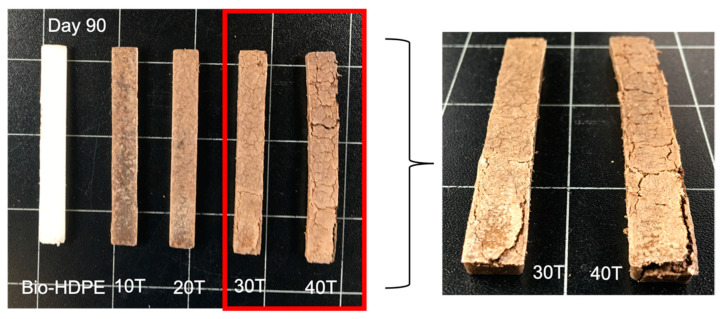
Visual appearance of disintegration of bio-based high-density polyethylene (Bio-HDPE) with different weight percentage of treated chia seed flour with APS under compost conditions.

**Table 1 polymers-13-02269-t001:** Weight composition of Bio-HDPE with chia seed flour (CSF) materials and labelling.

Reference	Parts by Weight (wt.%)		
	Bio-HDPE^1^	UTCSF^2^	TCSF^3^
Bio-HDPE	100	0	0
10T	90	0	10
20UT	80	20	0
20T	80	0	20
30T	70	0	30
40T	60	0	40

^1^ Bio-based high-density polyethylene; ^2^ Untreated chia seed flour; ^3^ Treated chia seed flour.

**Table 2 polymers-13-02269-t002:** Mechanical properties for Bio-HDPE with untreated and treated chia seed flour.

Reference	Tensile Strength (MPa)	Tensile Modulus (MPa)	Elongation at Break (%)	Flexural Strength (MPa)	Flexural Modulus (MPa)	Shore D Hardness	Impact Absorbed Energy (kJ·m^−2^)
Bio-HDPE ^1^	19.0 ± 1.2	373.0 ± 16.0	520.0 ± 18.5	23.9 ± 1.21	804.0 ± 38.0	56.6 ± 0.5	2.75 ± 0.2
20UT ^2^	10.7 ± 0.5	374.0 ± 14.7	35.9 ± 5.7	23.13 ± 0.76	784.9 ± 34.5	61.0 ± 0.7	1.67 ± 0.08
20T ^3^	12.0 ± 0.4	396.5 ± 14.5	50.8 ± 5.4	24.1 ± 0.5	839.8 ± 21.6	62.2 ± 1.0	1.88 ± 0.1

^1^ Bio-based high-density polyethylene; ^2^ 20 wt.% untreated chia seed flour; ^3^ 20 wt.% treated chia seed flour.

**Table 3 polymers-13-02269-t003:** Main thermal parameters of Bio-HDPE with untreated and treated chia seed flour obtained using DSC.

Reference	T_m_ (°C) ^1^	ΔH_m_ (J g^−1^) ^2^	*Xc* (%) ^3^
Bio-HDPE ^4^	131.0	154.2	55.8
20UT ^5^	131.8	105.8	49.5
20T ^6^	131.5	126.1	57.8

^1^ Melt temperature; ^2^ Melt enthalpy; ^3^ Degree of crystallization; ^4^ Bio-based high-density polyethylene; ^5^ 20 wt.% untreated chia seed flour; ^6^ 20 wt.% treated chia seed flour.

**Table 4 polymers-13-02269-t004:** Effect of weight percentage of treated chia seed flour with APS on Shore D Hardness and impact absorbed energy.

Reference	Shore D Hardness	Impact Absorbed Energy (kJ·m^−2^)
Bio-HDPE ^1^	56.6 ± 0.7	2.75 ± 0.20
10T ^2^	59.4 ± 0.9	1.92 ± 0.10
20T ^2^	61.0 ± 0.7	1.88 ± 0.08
30T ^2^	62.2 ± 1.1	1.65 ± 0.12
40T ^2^	63.0 ± 0.8	1.62 ± 0.10

^1^ Bio-based high-density polyethylene; ^2^ wt.% of treated chia seed flour.

**Table 5 polymers-13-02269-t005:** Main thermal parameters of Bio-HDPE with different weight percentage of treated chia seed flour with APS obtained using DSC.

Reference	T_m_ (°C) ^1^	ΔH_m_ (J g^−1^) ^2^	*Xc* (%) ^3^
Bio-HDPE ^4^	131.0	154.2	55.8
10T ^5^	131.8	151.4	59.2
20T ^5^	131.5	126.1	57.8
30T ^5^	130.8	105.7	51.5
40T ^5^	131.0	86.3	49.1

^1^ Melt temperature; ^2^ Melt enthalpy; ^3^ Degree of crystallization; ^4^ Bio-based high-density polyethylene; ^5^ wt.% of treated chia seed flour.

**Table 6 polymers-13-02269-t006:** Colour coordinates (L*, a*, b*) for different weight percentage of treated chia seed flour with APS.

Reference	L* ^1^	a* ^2^	b* ^3^	ΔE_ab_* ^4^
Bio-HDPE	71.3 ± 0.3	−2.6 ± 0.1	−2.68 ± 0.02	
10T ^5^	42.5 ± 0.1	4.15 ± 0.08	6.92 ± 0.1	31.1
20T ^5^	42.1 ± 0.2	4.89 ± 0.2	7.96 ± 0.1	32.0
30T ^5^	36.6 ± 0.1	4.24 ± 0.07	5.72 ± 0.09	36.3
40T ^5^	37.2 ± 0.1	4.65 ± 0.1	6.73 ± 0.08	36.1

^1^ Luminance; ^2^ Green to red; ^3^ Blue to yellow; ^4^ Colour change regarding neat bio-based high-density polyethylene (Bio-HDPE); ^5^ wt.% of treated chia seed flour.

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
