# Peer review of "Contribution to a Circular Economy Model: From Lignocellulosic Wastes from the Extraction of Vegetable Oils to the Development of a New Composite"

_polymers, 2021, doi:10.3390/polym13142269_

Round 1
Reviewer 1 Report
The manuscript by Fombuena and co-workers describes the use of waste chia seed flour (CSF) as dopant for HDPE.
Several tests have been made to compare doped and undoped bio-HDPE, including the use of a silane agent to improve the compatibility between the materials, and a concentration screening. The results are well described, however, there are some crucial points to be addressed:
- The data were compared with different, previously reported, materials. Nevertheless, these are mostly silane-based additives. This is not very helpful to compare the current results with state of the art HDPE dopants with desired performance.
- Indeed, from a technological point of view, only minor improvements were achieved, becoming many properties less interesting than HDPR itself. This hardly justify the use of CSF as innovative dopant.
- The degradation advantages are overestimated, being only CSF degraded, letting the composite to fragment faster, (likely yielding microplastics?), without being actually bio-degraded.
- The concept of circular economy, in turn, become also abused, since no real advantages can be seen: a silane is needed and recycling seems even more difficult.
- In the title, the material is better defined as composite, not as compound.
Minor points:
Why silanes should be hydrolyzed into silanols before the addition of CSF? Which is the acidic trigger? If this is the case, polysiloxanes should also form. A better discussion is needed about the chemical form of the APS.
The sentences “Bio-HDPE presented a high elongation at break of 520%, showing a high ductility properties as has been reported previously [44]. This value decreased to 36% and 50.8% values in 20UT and 20T samples, respectively. CSF treated with APS provided a lower decrease in elongation at break, specifically a 90.2% respect to 93.1% compared to Bio-HDPE.” May generate confusion, owing to different comparison terms.
“Silanes (maybe better silanization?) is one of the most employed method to improve the adhesion/interaction between polymer matrix and lignocellulosic fiber.”
“coordenate” at page 20.
Reviewer 2 Report
Dear Authors,
Thank you for the nice and well written manuscript. However, I would like to ask to make some minor changes to improve the quality of this intriguing paper:
1) could you please shorten the aim at the end of Introduction - in the present form it is too large and both parts of aims are mixed with the present research... Otherwise Introduction is very well written and easy understandable!
2) Materials and methods also commonly are Ok, but give please the Reference for methodology in the chapter 2.5.;
3) Results are interesting, but decipher, please, the abbreviations beneath the Table 1 and everywhere for the Tables and Figures, where you use them;
additionally, indicate what exactly is seen in the Figure 1 and Figure 8, and indicated by an arrows (not all are EM specialists and could understand the picture without an explanation...)
4) Conclusions earn my main objection, - please, remove an aim, the extra words, maximally shorten them and make more precise. (They are just descriptive now and remind the Result, what is not valid for such a nice work);
5) do you really need the Reference 79 from the previous century? Your References are so nowadays that this "old" source simply doesnt fit for this well-elaborated chapter...
Round 2
Reviewer 1 Report
The manuscript can now be accepted. All critical issues have been addressed.
As a final comment, I must say that the use of waste materials as fillers should be carefully evaluated and justified from all technological point of view. In a circular economy perspective, for example, a new material should not negatively impact on existing recycling processes.
A further recommendation: please check some sentences among those added, to improve English sentences (e.g. "not signs of changes has been recorded" and "not any sign of weight loss") and to avoid repetition of concepts (e.g. the broad comment on biodegradability).